# **USGS Experience with the Residual Absolutes Method**

E. William Worthington<sup>1</sup>, Jürgen Matzka<sup>2</sup>

<sup>1</sup>U. S. Geological Survey, Geomagnetism Program, PO Box 25046, MS 966, Denver, CO 80225, USA <sup>2</sup>GFZ German Research Centre for Geosciences, Telegrafenberg, 14473, Potsdam, Germany

Correspondence to: E. William Worthington (bworth@usgs.gov)

**Abstract.** The U. S. Geological Survey (USGS) Geomagnetism Program has developed and tested the Residual method of Absolutes, with the assistance of the Danish Technical University's (DTU) Geomagnetism Program. The computations of the absolute and baseline values are presented with improved calculations, such as the exact conversion from nanoTeslas (nT) to degrees. Three years of testing were performed at College Magnetic Observatory (CMO) to compare the Residual method with the Null method. Results show that the two methods compare very well with each other and both sets of baseline data were used to process the 2015 Definitive data. The Residual method is also being used at the Deadhorse Magnetic Observatory and will be implemented at the other USGS high latitude geomagnetic observatories in the summer of 2017.

This draft manuscript is distributed solely for purposes of scientific peer review. Its content is deliberative and predecisional,
so it must not be disclosed or released by reviewers. Because the manuscript has not yet been approved for publication by the
U.S. Geological Survey (USGS), it does not represent any official USGS finding or policy.

## **1** Introduction


Geomagnetic observatories are unique facilities. They measure the variation of the three vector components of the geomagnetic field at either one-minute and/or one-second time resolution and they also measure the absolute value of the

- geomagnetic field (see, e.g. Rasson et al., 2007, Matzka et al, 2010, Love and Chulliat, 2013, Chulliat et al, 2016). The vector components are typically measured with a three-axis fluxgate magnetometer. The absolute measurements are used to generate baseline values, which are the difference between the absolute values and raw variation data, for each magnetic component. The baseline values are used to calibrate the variation data to produce final definitive data. Since the first systematic geomagnetic observations in the 16<sup>th</sup> century (Malin, 1987) there has been a continued development and improvement of
- instruments to measure absolute values of the geomagnetic field. Some of these instruments measure the strength of a magnetic vector component, the magnitude of the entire vector, or the angles of the orientation of the geomagnetic vector. Beginning in the 1970's the most common instruments employed are the proton precession magnetometer that measures the magnitude (F) of the geomagnetic vector, and since the 1980's, the Declination-Inclination Magnetometer (DIM), also known as the DI-Fluxgate, which is used to measure the declination (D) and inclination (I) angles of the geomagnetic vector. These three
- measurements make it possible to compute all other components of the vector (Jankowski and Sucksdorff, 1996). The DIM is used to measure four declination angles, in four different orientations, and four inclination angles. This instrument is used as

a null detector, where the output of the fluxgate sensor is zero when the magnetic field vector is perpendicular to the sensor. The so-called "Null method" was the technique first developed for use with the DIM. This method requires the observer to rotate the instrument so the analog output of the fluxgate, in nT, is nulled and the time of the null reading and the angle are recorded. The Null method works well and is still in use today. The angular readings are read through a microscope where,

- for the Zeiss 020 theodolite discussed here, the angle is read to degrees, minutes of arc, and estimated to the tenth of a minute. However, there are two drawbacks to the Null method. 1) At high latitudes, where the geomagnetic field is more active, a good null can be difficult to obtain. 2) The Null method requires the observer to be within arm's reach of the DIM to constantly adjust the theodolite/sensor orientation to achieve a nulled output on the fluxgate. If the observer is not free of ferrous materials, such as watches, keys, tools, dental work, or small electronics, then the presence of these items can contaminate the
- measurements.

The point of the Residual method is to allow readings of the horizontal or vertical circle for positions where the output of the fluxgate, termed the residual value in nT, is not exactly zero. This makes it easier to cope with rapid changes of the geomagnetic field. It also allows the observer to be farther away from the DIM, reducing the possibility of contaminated

- measurements. Additionally, as in the case for Zeiss 020 theodolite, the circle reading can be set exactly to a whole minute value and the output of the DIM magnetometer can be used to mathematically compensate for the resulting small deviation in angle between the whole minutes as opposed to estimating tenths of minutes by eye. The Residual method presented here was developed for the Danish geomagnetic observatories at Danish Technical University (DTU), originally part of the Danish Meteorological Institute. The method and computations are based on a document written by E. Kring Lauridsen (1985), the
- book by Jankowski and Sucksdorff (1996) and the study by Matzka and Hansen (2007). The computations shown in this paper have been updated to increase the accuracy and precision of the measurements.

This paper discusses the updated computational scheme adopted by the USGS and shows comparisons of the two methods employed at the USGS College Magnetic Observatory.



# 2.0 Computations

For the sake of simplicity, all of the following equations will use angles in radians. For programing purposes, angles measured in degrees or gradians will need to be converted to radians as appropriate. Instrument orientations, such as West Down, refer to the direction the telescope is pointing and the position of the fluxgate sensor mounted on the telescope. Before going into detail about the computations, the following definitions will be used.

 $nT = Value of magnetic field strength, 1 nT = 10^{-9} Teslas$ 

|    | $\mathbf{H} = Absolute value of the horizontal intensity, in nT.$                                          |
|----|------------------------------------------------------------------------------------------------------------|
|    | $\mathbf{D}$ = Absolute value of the declination angle, in minutes.                                        |
|    | $\mathbf{Z}$ = Absolute value of the vertical intensity, in nT.                                            |
|    | $\mathbf{F}$ = Absolute magnitude of the magnetic field vector in nT, also known as the Total Field.       |
| 5  | $\mathbf{I}$ = Absolute value of the inclination angle, in degrees.                                        |
|    | E = Declination value in nT, recorded by the observatory fluxgate                                          |
|    | $h_i$ = the ith variation value for the horizontal field, H, from the fluxgate, in nT.                     |
|    | $e_i$ = the ith variation value for the declination field, E, from the fluxgate, in nT.                    |
|    | $z_i$ = the ith variation value for the vertical field, Z, from the fluxgate, in nT.                       |
| 10 | $f_i$ = the ith variation value for the total field, from the overhauser, in nT.                           |
|    | $R_i$ = the ith residual value, in nT, i=1-4 for D readings, i=5-9, for I readings                         |
|    | $AD_i$ = the ith measured declination angle in decimal degrees                                             |
|    | $A_1$ = computed angle for West Down, in decimal degrees                                                   |
| 15 | $A_2$ = computed angle for East Down, in decimal degrees                                                   |
|    | $A_3$ = computed angle for West Up, in decimal degrees                                                     |
|    | $A_4$ = computed angle for East Up, in decimal degrees                                                     |
|    | HS = hemisphere, 1 for northern, -1 for southern                                                           |
|    | M = magnetic meridian angle, mean of A <sub>1</sub> , A <sub>2</sub> , A <sub>3</sub> , and A <sub>4</sub> |
| 20 |                                                                                                            |
|    | $AI_i$ = the ith measured inclination angle in decimal degrees                                             |
|    | $I_1$ = computed angle for South Down, in decimal degrees                                                  |
|    | $I_2$ = computed angle for North Up, in decimal degrees                                                    |
|    | $I_3$ = computed angle for South Up, in decimal degrees                                                    |
| 25 | $I_4$ = computed angle for North Down, in decimal degrees                                                  |
|    | $I_5$ = computed calibration angle for South Up, in decimal degrees                                        |
|    | $H_b = baseline value for H, in nT$                                                                        |
|    | $Z_{\rm b}$ = baseline value for Z, in nT                                                                  |
| 30 | $D_b$ = baseline value for D, in minutes                                                                   |
|    | $F_{pc} = F$ pier correction, between the total field instrument and absolute pier, in nT                  |
|    | $MU_1$ = First Mark Up reading, in decimal degrees                                                         |
|    | $MD_1 =$ First Mark Down reading, in decimal degrees                                                       |
|    | $MU_2$ = Second Mark Up reading, in decimal degrees                                                        |

 $MD_2$  = Second Mark Down reading, in decimal degrees

- MA = mean value of the four Mark readings
- AZ = True azimuth angle to the Azimuth mark, in decimal degrees
- SV = Scale value of the fluxgate used on the DIM output

5

# **2.1 Inclination Computations**

The inclination angle computations are discussed first because the Horizontal baseline value,  $H_b$ , is needed for the computation 10 of D. For reference, using the null method, the computation of the Inclination angle is fairly simple,

$$\mathbf{I_N} = \frac{(AI_1 + AI_2) - (AI_3 + AI_4) + 360}{4}.$$
 (1)

 $AI_{1-4}$  refers to the South Down orientation, the North Up orientation, the South Up orientation, and the North Down orientation respectively. This sequence is the USGS order for the four inclination measurements.

15

20

For the Residual method, the computations involve some additional steps. Each inclination reading (AI<sub>i</sub>) is first corrected using the corresponding residual value.

$$I_1 = AI_1 + HS \,\sin^{-1}\left(\frac{R_5}{f_5}\right) - 180 \tag{2}$$

$$I_2 = AI_2 - HS \,\sin^{-1}\left(\frac{R_6}{f_6}\right) \tag{3}$$

$$I_3 = 180 - \left(AI_3 - HS \sin^{-1}\left(\frac{R_7}{f_7}\right)\right)$$
(4)

$$I_4 = 360 - \left(AI_4 + HS \sin^{-1}\left(\frac{R_8}{f_8}\right)\right)$$
(5)

The inclination is first computed by taking the mean of these four angles.

$$\mathbf{I}_{\text{mean}} = \frac{(\mathbf{I}_1 + \mathbf{I}_2 + \mathbf{I}_3 + \mathbf{I}_4)}{4} \tag{6}$$

However, the computation can be further improved by making a correction to the individual F values in the computation of  $I_1$  to  $I_4$ . The individual values for H, E, and Z are used to correct for changes in the orientation of F in the computation.

The individual F values  $f_5$  through  $f_9$  are corrected by

$$f_{i} = F_{mean} + (h_{i} - h_{mean}) \cos I_{mean} + (z_{i} - z_{mean}) \sin I_{mean} + \frac{(e_{i}^{2} - e_{mean}^{2})}{2 F_{mean}}$$
(7)

Where

$$h_{mean} = \frac{(h_5 + h_6 + h_7 + h_8 + h_9)}{5},\tag{8}$$

$$e_{mean} = \frac{(e_5 + e_6 + e_7 + e_8 + e_9)}{5},\tag{9}$$


$$z_{mean} = \frac{(z_5 + z_6 + z_7 + z_8 + z_9)}{5}.$$
 (10)

$$F_{mean} = \frac{(f_5 + f_6 + f_7 + f_8 + f_9)}{5},\tag{11}$$

This correction introduces an iteration to the computation for I. After the various values of I are computed using equations 2 through 6,  $I_{mean}$  is used in equation 7 to correct the values for  $f_5$ ,  $f_6$ ,  $f_7$ ,  $f_8$ , and  $f_9$ . I is then computed again using equations 2 through 6. This iteration are constitued antil the shares in  $I_{mean}$  is used in equation 7 to correct the values for  $f_5$ ,  $f_6$ ,  $f_7$ ,  $f_8$ , and  $f_9$ . I is then computed again using equations 2 through 6. This iteration are constitued antil the shares in  $I_{mean}$  is used in equation 7 to correct the values for  $f_5$ ,  $f_6$ ,  $f_7$ ,  $f_8$ , and  $f_9$ . I is then computed again using equations 2 through 6.

through 6. This iterative process is continued until the change in  $I_{mean}$  from one iteration to the next is less than 0.0001 degrees.

The final inclination value I is used to compute the absolute values for the H and Z. To obtain a final value for F, it is necessary to add in the F pier correction,  $F_{pc}$ ,

$$\mathbf{F} = F_{mean} + F_{pc} \tag{12}$$


Then **H** and **Z**, for the times of the four Inclination measurements can be computed using **F** and **I**,

$$\mathbf{H} = \mathbf{F} \, \cos \mathbf{I} \tag{13}$$

and

$$\mathbf{Z} = \mathbf{F} \, \sin \mathbf{I} \tag{14}$$

These two absolute values can be used to compute the baseline values for H and Z.

$$H_b = \sqrt{\mathbf{H}^2 - \mathbf{E}_{mean}^2} - H_{mean} \tag{15}$$

$$Z_b = \mathbf{Z} - Z_{mean} \tag{16}$$


An extra or fifth inclination measurement is performed to determine the scale value of the fluxgate magnetometer mounted on the DIM. This is used to verify the fluxgate's scale value. The angular difference of the telescope between the  $4^{th}$  and  $5^{th}$  I readings (elements 8 and 9, for h and z, respectively) is exactly 10.0 minutes, or 0.16667 degrees (for a theodolite that measures

in gradians (gon) one would use 0.2 gon). The angular change of the fluxgate with respect to the magnetic field (from perpendicular to slightly tilted) is denoted  $\Delta B$  (Eqn. 17). The change in residuals is denoted as  $\Delta R$ . Computation of the scale value thus follows:

$$\Delta B = 0.16667 + \left(-\sin\mathbf{I} * \frac{(h_9 - h_8)}{\mathbf{F}} + \cos\mathbf{I} * \frac{(z_9 - z_8)}{\mathbf{F}}\right) \frac{180}{\pi}$$
(17)

$$\Delta R = R_9 - R_8 \tag{18}$$

$$SV = \mathbf{F} \frac{\Delta B}{\Delta R} \tag{19}$$


Ideally the resulting scale value should be 1.000, indicating that the output of the DIM fluxgate is in fact nT. In practice, it can range from 0.99 to 1.01. As long as the measured residual values are within  $\pm 10.0$  nT there is no need to adjust for a change in the scale value. Monitoring and tracking the scale value can help with diagnosing any problem that might develop with the instrument.

5

10

# **2.2 Declination Computations**

Computing the absolute value for declination, using the Null method, from the declination measurements is fairly simple. The magnetic meridian is computed as the average of the four declination readings. The four mark readings, which are sightings on the True Azimuth mark before and after the declination measurements, are also averaged. The computation of the absolute value for declination, using the Null method, can be described in simple terms as:

 $\mathbf{D}_{N}$  = Magnetic meridian – average mark readings + the True Azimuth of the mark.

The baseline value is easily computed by taking the difference between  $\mathbf{D}_N$  and the value for declination from the fluxgate. 15 However, this is complicated by the fact that the output of the fluxgate is in nT (E) and must be converted to an angular value. The conversion traditionally used is known as the small angle approximation. More detail on this can be found in Jankowski and Sucksdorff (1996) and Wienert (1970).

The exact formula for the Declination conversion uses the simple trigonometric relation that the Declination angle can be computed from the inverse tangent of the value of E divided by the absolute value of H. Therefore the calculations are more complex than those for the Null method. The ordinate or magnetometer values, from the observatory fluxgate for  $e_{1-4}$  are converted to an angle using the exact conversion as:

$$d_i = \tan^{-1} \left( \frac{e_i}{(h_i + H_b)} \right) \tag{20}$$

With advances in computing power the small angle approximation is no longer necessary and this exact formula is easily computed. Using equation 20 can provide a more precise angular value for declination when converting the value from E.

The USGS declination measurements are in the following order: West Down, East Down, West Up, and East Up which correspond to  $AD_1$  through  $AD_4$  respectively. The angles for each reading are computed in the following fashion,

$$A_1 = AD_1 - \sin^{-1} \left( \frac{R_1}{\sqrt{(h_1 + H_b)^2 + e_1^2}} \right) - d_1$$
(21)

$$A_2 = AD_2 + \sin^{-1} \left( \frac{R_2}{\sqrt{(h_2 + H_b)^2 + e_2^2}} \right) - d_2$$
(22)

$$A_3 = AD_3 - \sin^{-1}\left(\frac{R_3}{\sqrt{(h_3 + H_b)^2 + e_3^2}}\right) - d_3$$
(23)

$$A_4 = AD_4 + \sin^{-1} \left( \frac{R_4}{\sqrt{(h_4 + H_b)^2 + e_4^2}} \right) - d_4 \tag{24}$$



Equations 21 - 24 are more complex than those used for the Null method, because they have two terms that add to the accuracy of the computations. The term inside the inverse sine function, the residual (R<sub>i</sub>) divided by H, represents the interpolated value added to the angular measurement (AD<sub>i</sub>). The interpolation value corrects for the H baseline value and the variation about the orientation axis (H) of the fluxgate sensor. The second term represents the value of ei for each measurement, converted to an

angle, as computed by equation 20.

The mean of the four angles A<sub>i</sub>, termed the Magnetic Meridian, is next computed,

$$M = \frac{(A_1 + A_2 + A_3 + A_4)}{4}.$$
 (25)


Similarly, the mean should be computed for the four mark angles,

$$MA = \frac{(MU_1 + MU_2 + MD_1 + MD_2)}{4}.$$
 (26)

The baseline value for Declination, in minutes, is computed, as follows,

$$D_{b} = (M - MA + AZ) \cdot 60 \tag{27}$$

Where AZ is the true azimuth angle to the Azimuth mark, in decimal degrees.

#### 2.3 Final Absolute Values

In this method, the absolute values for **H**, **D**, and **Z** are computed for the starting time of the measurements, corresponding to the West Down measurement, for use in data processing. These final absolute values are denoted with a subscripted **0** to avoid confusion. So the value for H would be determined by the following:

$$\mathbf{H_0} = \sqrt{(\mathbf{h_1} + \mathbf{H_b})^2 + \mathbf{e}_1^2}.$$
 (28)

The value for **D** is computed as:

$$\mathbf{D_0} = \mathbf{D_b} + \tan^{-1} \left( \frac{\mathbf{e_1}}{\mathbf{h_1} + \mathbf{H_b}} \right) \frac{180}{\pi}.$$
 (29)

The absolute value for Z would be,

$$\mathbf{Z}_0 = \mathbf{Z}_b + \mathbf{z}_1. \tag{29}$$

#### 15 2.4 Diagnostic Fluxgate Parameters

There are five separate error parameters that can be computed from the measured declination and inclination angles that are useful for diagnosing the quality of measurements performed with the DIM.

For the D measurements, there are the two angles, also known as the collimation error or misalignment angles, also termed the Sight Error and the Azimuth Error by Rasson (2005).

The Declination Sight Error, in seconds of arc, designated as  $\varepsilon_D$  is computed as follows:

$$\varepsilon_D = \frac{(A_4 + A_3 - A_2 - A_1)}{4 \, \tan \mathbf{I}} \, 3600 \,. \tag{30}$$

The Azimuth Error, in seconds of arc, designated as  $\delta$ , is computed as follows:

$$\delta = \frac{(A_4 - A_3 - A_2 + A_1)}{4} \ 3600. \tag{31}$$

The Declination sensor offset, defined as the sensor reading in a true zero magnetic field, also called Sensor Magnetization Error by Rasson (2005), designated as SO<sub>D</sub>, is computed in nT,

$$SO_D = \mathbf{H} \left(\frac{\pi}{180}\right) \frac{(A_4 - A_3 + A_2 - A_1)}{4}.$$
 (32)

The Sight Error and Sensor Offset can also be computed for the Inclination readings, whereas the Azimuth error cannot be determined for inclination.

The Inclination Sight Error, in seconds of arc, designated as  $\varepsilon_l$  is computed as follows:

$$\varepsilon_I = \frac{(I_2 + I_1 - I_4 - I_3)}{4} \ 3600. \tag{33}$$

The Inclination Sensor Magnetization Error, designated as SOI, is also computed in nT,

$$SO_{I} = -\mathbf{F} \left(\frac{\pi}{180}\right) \frac{(l_{2} - l_{1} - l_{4} + l_{3})}{4}.$$
 (34)

- 10 These error parameters have two uses. In the first case, these error values are computed for an individual set, and can be compared to the values of previous or following sets. The resulting values should be approximately the same. If one value is noticeably different it usually indicates a bad set of observations. In the second case, these error values can be tracked over longer periods to see if there are any drifts or changes in the results. Long term changes can indicate developing problems with the instrument, including contamination, a loose sensor, or other mechanical problems with the theodolite.
- 15

5

## **3.0 Absolute Measurement Tests**

The Residual method was tested at most of the USGS observatories, by USGS staff during site visits over the course of a year, and the agreement between the Null and Residual methods was satisfactory. More extensive testing was performed at the

- 20 College Magnetic Observatory (CMO). This was a logical choice because the observatory has good baseline stability and is located at high geomagnetic latitude, 65° N, allowing us to test the Residual method at high latitude. In addition, there were two observers performing absolutes three times a week. One of the observers was trained to use the Residual method once a week while the other observer continued using the null method twice a week. After six months, the second observer was trained to use the Residual method so that both observers could alternate methods to eliminate the possibility of an observer
- 25 bias, which could otherwise take months to identify. This overlap of techniques was started in mid-2012 and still continues. Baseline results of these tests are shown in Figure 1.