# Peer review of "USGS Experience with the Residual Absolutes Method"

_Geoscientific Instrumentation, Methods and Data Systems, 2017_

## Referee Comment (RC1) · L.ÂăW. Pedersen (Referee) · 29 May 2017

**USGS residual abs. Method.**

All in all a very good and precise manuscript, so I have rather few comments:

Corrections:
Page 3, line 26: fifth I-position must be North Down, same as 4. position, only slightly tilted.
Page 10, line 25: year must be mid-2015, not 2012.

Discussion:
Page 5:
How much does the inclination result improve using equation 7 to 11 instead of using only equation 6? Using the iteration process you end up with an error of about 0.0001 degree, but what is the error if you only uses equation 6?

Page 7:
Again, how much better is it to use the exact formula (equation 20) for the declination instead of the small angle approximation?
My point is again that it would be nice to get a feeling, if it can be worth to enhance the absolute calculations on a certain observatory. Will it improve the results on all locations or mostly at high lattitude ?

Page 6, line14.
Calculation the scale value of the DIM: Can you explain why you move the theodolite  10.0 minuts, 0.16667 degrees or 0.2 gon between 4. and 5. I reading? I know that it is due to the reading on the telescope, but an explanation could be nice to have.

Page 9:
Diagnostic fluxgate parameters:
The two misalignment angles that can be calculated are the misalignment of the fluxgate sensor in horizontal and vertical plane, and it is my opinion that they should be called horizontal misalignment and vertical misalignment.
The two sensor offsets $SO_D$ and $SO_I$ calculated from D and I readings should end up with the same value, so they can be compared for each set of measurements. I would prefer to call SO for 'sensor offset' even the offset are a combination of offset in sensor, in electronics and in cable. But using the same setup everytime (same sensor, cable and electronics), it should be a rather constant value.

The mentioned DIM scale value on page 6 is a 6. parameter that is good to follow.
Also the angles of the declination mark is good to follow over time to see if the theodolite pillar is rotating or the horizontal scale in the teodolite is loose.

I agree that it is essential to follow these parameters, and I would like to see a plot of at least some of them in the article.

4.0 Discussion and conclusion, page 11 and 12:
One of the advantages of using the residual method is, as mentioned, that the observer can be several meters away from the theodolite and will not influence the magnetic field as much, if he is magnetic. This is a very important 'feature' for us using glasses.

But another important improvement using the residual method, is the timing. It is easier to read the residual value at a certain time, than try to keep residual at zero and read the time.

---

## Referee Comment (RC2) · Anonymous Referee #2 · 1 Jun 2017

**USGS Experience with the Residual Absolutes Method**

by E. W.Worthington & J. Matzka

This paper presents a precise calculation of baseline values (discrete values) from Declination/Inclination measurements by the residual method.

The article is well organized and notations in equations are well described. The six following comments are mainly dealing with the fact that the paper lacks of examples, statistics and evidences onto the improvements of the residuals method. Then, one main outcome would be the possibility to provide estimations of error bars on each punctual absolute measurement.
 (That would be possibly used in subsequent baseline calculations -not necessarily smooth ones- with the goal of providing users with estimates of the data error standard deviations. But that very last point is beyond the scope of this paper.)

**COMMENTS**

**A- *Abstract*, page 1, lines 11 and 12:**
*"The Residual method is also being used at the Deadhorse Magnetic Observatory and will be implemented at the other USGS high latitude geomagnetic observatories in the summer of 2017."*
This last sentence appears useless here as only CMO baselines over 14 months are presented.
Moreover, explanations presented page 11, lines 21-22 *("The Residual method has been implemented at both College and Deadhorse observatories. The USGS plans to implement the Residual method at the remaining USGS observatories in Alaska in 2017 and in all observatories by 2019. "*) appear sufficient unless the authors add substantial new graphical and statistical examples with Deadhorse and observatories in Alaska.

**B- Section *1 Introduction*, page 2, line 6 to 8:**
*"1) At high latitudes, where the geomagnetic field is more active, a good null can be difficult to obtain. 2) The Null method requires the observer to be within arm's reach of the DIM to constantly adjust the theodolite/sensor orientation to achieve a nulled output on the fluxgate."*
Point 3) would be the fact that the residual method is far easier to teach and to be achieved by "observers" who are just on site for a year. Indeed, in some places, magnetic observatory observers are just people that will take care of magnetic measurements without knowing anything about geomagnetism or even about geosciences. Thus, the training has to be fast, easy, efficient and to allow possible clumsiness (as forgotten watch, magnetic glasses, etc).

**C- Section *2.0 Computations*, page 5, line 15:**

*"This iterative process is continued until the change in Imean from one iteration to the next is less than 0.0001 degrees."*

It makes sense that the iteration is stopped when reaching the limit of angle reading on the theodolite.

However, how is the iterative calculation (from equations 7 to 11) improving the calculation of Inclination? Please, show the distribution of Inclination values with "usual simple method" and the new exact calculation you propose. Please, convince the reader that the exact calculation is worthwhile.

**D- Section 2.2 Declination computations, page 7, line 25:**

*"Using equation 20 can provide a more precise angular value for declination when converting the value from E."*

Same question as for **C**-, Please, show statistical evidence for improvement of declination calculation.

**E- Section 2.4 Diagnostic Fluxgate Parameters, page 9 and 10.**

Please, compare and show the evolution over time of error parameters for both methods ("commonly used" and new exact calculation).

**F-** Getting precise discrete baseline values is only worthwhile if the discrete data are subsequently incorporated into a precise and mathematically exact determination of the baselines (rather than spline method on each baseline separately) such as, for example, the one presented during the last IAGA workshop in Dourbes:

> Lesur V., Heumez B., Telali A. and Coïsson P. (2016), "On the accuracy of CLF observatory data", In Proceedings of the XVIIth International Association of Geomagnetism and Aeronomy (IAGA) workshop.
> Or
> Lesur V., Heumez B., Telali A., Lalanne X., and Soloviev A. (2017) "Estimating error statistics for Chambon-la-Forêt observatory definitive data" [Paper under review for (probably) the same journal as the present paper].

This raises the issue of obtaining error estimates on observatory measurements.

How the authors may give error bar estimations on each punctual absolute measurement?

(The reviewer precise that he is not co-author or even associated in any sense to the above-mentioned paper.)

---

## Author Comment (AC1) · 14 Jul 2017

The first correction was made. The second was not. He probably mis-understood the sentence.

Discussion

Page 5: It turns out that this is actually very small except in cases where F and I are not measured simultaneously. We have left it in for completeness. A further explanation has been added to the text.

Page 7: This comment was addressed with Figure 1 and further explanation in the text.

Page 6: Thus was addressed with a further explanation.

[Figure]

Page 9: We chose not to change the terminology at this point. A graph, demonstrating the utility of the diagnostic parameters, was added in Figure 2.

The two comments for section 4.0 were both added to the text.

The original Figure 1, is now Figure 3.
* * *
[Figure]

**Fig. 1.**

[Figure]

[Figure]

**Fig. 2.**

[Figure]

[Figure]

---

## Author Comment (AC2) · 14 Jul 2017

Comment A was addressed and the phrase about Deadhorse was removed.

Comment B. The suggested point 3 is addressed in the conclusions.

Comment C is the same as Pedersen's page 5 comment and has been addressed.

Comment D is the same as Pedersen's page 7 comment and has been addressed. Figure 1. and further text was added to address this.

Comment E is similar to Pedersen's page 9 comment. We presented the parameters for the residual method (Figure 2) but we do not compute the parameters for the null method.

[Figure]

Comment F. This is a very important comment, however we feel it is beyond the scope of this paper and should be addressed separately.